# Adipose Tissue-Derived Components: From Cells to Tissue Glue to Treat Dermal Damage

**DOI:** 10.3390/bioengineering10030328

**Published:** 2023-03-05

**Authors:** Linda Vriend, Berend van der Lei, Martin C. Harmsen, Joris A. van Dongen

**Affiliations:** 1Department of Plastic Surgery, University of Utrecht, University Medical Center Utrecht, 3584 CS Utrecht, The Netherlands; 2Department of Pathology & Medical Biology, University of Groningen, University Medical Center Groningen, 9700 AC Groningen, The Netherlands; 3Department of Plastic Surgery, University of Groningen, University Medical Center Groningen, 9700 AC Groningen, The Netherlands; 4Bergman Clinics, 8443 CG Heerenveen, The Netherlands; 5Bergman Clinics, 2289 CM Rijswijk, The Netherlands

**Keywords:** dermal damage, ASCs, ASC secretome, SVF, ECM, Extracellular matrix, hydrogels, native hydrogels, acellular hydrogels, wound healing

## Abstract

In recent decades, adipose tissue transplantation has become an essential treatment modality for tissue (volume) restoration and regeneration. The regenerative application of adipose tissue has only recently proven its usefulness; for example, the method is useful in reducing dermal scarring and accelerating skin-wound healing. The therapeutic effect is ascribed to the tissue stromal vascular fraction (tSVF) in adipose tissue. This consists of stromal cells, the trophic factors they secrete and the extracellular matrix (ECM), which have immune-modulating, pro-angiogenic and anti-fibrotic properties. This concise review focused on dermal regeneration using the following adipose-tissue components: adipose-tissue-derived stromal cells (ASCs), their secreted trophic factors (ASCs secretome), and the ECM. The opportunities of using a therapeutically functional scaffold, composed of a decellularized ECM hydrogel loaded with trophic factors of ASCs, to enhance wound healing are explored as well. An ECM-based hydrogel loaded with trophic factors combines all regenerative components of adipose tissue, while averting the possible disadvantages of the therapeutic use of adipose tissue, e.g., the necessity of liposuction procedures with a (small) risk of complications, the impossibility of interpatient use, and the limited storage options.

## 1. Introduction

Soft tissue injury is a kickstart for the wound-healing cascade. During the course of wound healing, resident cells, together with invading immune cells, rebuild tissues and restore their function to a pre-injured state. This comprises four essential and overlapping phases: hemostasis, inflammation, proliferation, and remodeling [1,2]. Systemic and local microenvironmental conditions may dysregulate and compromise adequate wound healing. The bacterial infection of wounds negatively impacts wound healing. After the initial phagocytosis of invading granulocytes, macrophages continue to eradicate both bacteria and cell debris, e.g., dead granulocytes. This requires pro-inflammatory M1-type macrophages. During smoldering infections, when M1-type macrophages are prolongedly present and activated, the proper accumulation of pro-regenerative M2-type macrophages may be delayed or prevented, which can compromise wound healing [3]. Diabetic ulcers at the lower extremities, which are a result of polyneuropathy, periphery vascular disease, and chronically infected wounds, are exemplary in this respect. The extended inflammatory phase that causes inadequate wound healing also prolongs the presence of Transforming Growth Factor-beta (TGF-b). This promotes the differentiation of fibroblasts into myofibroblasts and causes the aberrant deposition of highly crosslinked extracellular matrix (ECM) components such as collagens. The local accumulation of crosslinked ECM equates to fibrotic scarring. Not only does growth factor stimulation of (myo)fibroblasts cause scarring, but the high stiffness of scars stimulates (myo)fibroblasts via mechanosignaling [4]. Both high stiffness and activated (myo)fibroblasts add to the maintenance of the pro-fibrotic process, leading to scarring.

In pathological (hypertrophic) scarring, the skin’s texture is uneven, less elastic and pliable, and the color is either erythematous or the surface can be hypopigmented or hyperpigmented compared to normal skin. Thus, the scarring of the skin will lead to both functionally and aesthetically displeasing outcomes after dermal injury. In search of new treatment modalities to augment skin wound healing and to prevent or reverse dermal scarring, the use of adipose tissue (lipofilling or fat grafting) has become a hot topic of interest, both in daily reconstructive plastic surgery as well as in regenerative medicine research [5,6,7,8,9,10,11,12]. Several clinical studies have reported improved skin quality, faster wound healing, a decrease in functional scar-restrictions, and the alleviation of scar-related pain after fat grafting [5,6,7,8,9,10,11,12]

This concise review discusses the regenerative components of adipose tissue, i.e., the stromal vascular fraction (SVF), adipose tissue-derived stromal cells (ASCs) and their secretome, and the ECM, and their therapeutic use to treat dermal damage on a cellular level. Finally, future directions for adipose-tissue-derived therapy are suggested.

## 2. Regenerative Components of Adipose Tissue

Adipose tissue basically consists of parenchyma (adipocytes), stroma, and ECM. The regenerative potency of adipose tissue is mostly attributable to the 10% (*v*/*v*) non-parenchymal content that comprises the SVF [13]. The stromal cells comprise vessels, including endothelial cells, vascular smooth muscle cells, pericytes, and mesenchymal stromal cells, which are mainly fibroblasts, pre-adipocytes, and other mesenchymal precursor cells [14,15]. Of the precursor cells, a part develops into ASCs when cultured. Culture-derived ASCs are multipotent cells that can differentiate into ectodermal [16], endodermal [17], and mesodermal lineages [18] and have a proven therapeutic capability [19]. The precursor cells of ASCs mostly reside in living SVF. Yet, little is known about whether their regenerative capability is similar to that of cultured ASCs.

SVF and ASCs are promising as treatment modalities in, for example, dermal damage, specifically through the immune modulating, anti-fibrotic, and remodeling properties that they hold. ASCs can mediate inflammatory responses by modulating inflammation through the recruitment of (M2) macrophages that secrete the immune suppressive interleukin-10 (IL-10) [20]. These M2 macrophages control tissue regeneration and ECM remodeling. This alters the balance of ECM degradation and matrix deposition. For this, M2 macrophages release ECM-degrading proteases such as MMPs while they modulate ECM deposition. ASCs inhibit M1 macrophages and stimulate their repolarization to M2 macrophages, and both contribute to a pro-regenerative shift [21] Additionally, ECM also modulates the immune response through the embedding of signaling molecules such as matrikines that are released after partial ECM proteolysis [22]. Moreover, depending on the composition and stiffness of ECM in tSVF, ECM may facilitate the transition of M1 to M2 phenotypes of macrophages [23]. This, in turn, stimulates progenitor cell chemotaxis, cell proliferation and differentiation, and a shorter pro-inflammatory effect [24,25]. To illustrate the point, tSVF injection accelerated primary dermal wound healing (healing by primary intention after the approximation of two injured wound edges) and improved scar appearance after six months of follow-up compared to baseline [26]. This observation of the prevention of scarring effect was temporary, indicating the normal course of wound healing had matched the effect of tSVF at the end of follow-up (twelve months). In fact, this suggests that tSVF boosted primary wound healing more than it delayed scarring by exerting its anti-inflammatory effect and that it improved matrix remodeling effect, as described above.

tSVF may also exert remodeling and anti-fibrotic properties by influencing myofibroblast differentiation and ECM deposition, modification, and degradation. During ECM remodeling, the early deposited ECM is modified. In the beginning, mostly collagen III is deposited, as is collagen I in smaller amounts. At the end of wound healing, the balance between collagen III and I shifts, resulting in the dominance of the stronger collagen I [27,28]. A balance between collagen production and collagen degradation and the presence of collagen I are key to preventing scarring. ASC secretes factors including fibroblast growth factor 1 (FGF1), FGF2, hepatocyte growth factor (HGF), and insulin-like growth factor-1 (IGF-1), which are anti-fibrotic and play a part in maintaining this balance [29,30]. The severity of scarring is also determined by the degree of collagen crosslinking. A disbalance between collagen deposition and crosslinking versus the degradation by matrix metalloproteinases (MMPs), crosslinking enzymes such as procollagen-lysine, 2-oxoglutarate 5-dioxygenases (PLODs), and other matrix-degrading proteases results in a net accumulation of heavily crosslinked ECM [31]. Interestingly, ASCs secrete MMP2 and MMP9, which may facilitate ECM degradation [32].

Furthermore, it is known that TGF-b acts in a concentration-dependent fashion. At low concentrations, TGF-b promotes angiogenesis and inhibits inflammation. In contrast, at high concentrations, TGF-b acts in a chronic pro-fibrotic fashion. In vitro research showed that dermal fibroblasts, stimulated with TGF-b, resulted in less proliferation when treated with ASC-secreted trophic factors. Moreover, trophic factors released by ASCs also prevented continuous differentiation into myofibroblasts and the excessive deposition of ECM [32]. As tSVF comprises ASCs, it is possible that tSVF may have similar effects. In the normal course of wound healing, TGF-b drives fibroblast differentiation into myofibroblasts to contract the wound. Myofibroblasts and fibroblasts deposit collagen types I and III and other ECM proteins, which replace the granulation tissue that is formed in the inflammation phase [33]. As a result, there is a dominance of the stronger collagen type I. Myofibroblasts have a contractile and more migratory phenotype than fibroblasts. Under the influence of TGF-b, large numbers of myofibroblasts and their prolonged activation result in the formation of a chronic pro-fibrotic stimulus through the excessive secretion of ECM, which induces fibrosis and thicker or contractile (hypertrophic) scarring [32,33,34]. The secreted trophic factors from ASCs could thus prevent the excessive deposition of ECM by myofibroblasts, thereby contributing to the prevention of scarring.

The clinical practice of cell-based therapy, including ASC therapy, has limitations. In general, preoperative liposuction risks are low and postoperative complications are minor and infrequent (wound infection, ecchymosis). However, liposuction procedures pose an increased risk in patients with cardiovascular disease, morbid obesity, diabetes, or multiple comorbidities. Although surgeons usually refrain from operating on patients with multiple comorbidities, especially in morbidly obese and smoking patients, the popularity and demand of liposuction procedures have increased in this patient category and, with that, a few reports of more serious complications such as visceral perforation following liposuction have also accumulated [35]. Moreover, intra-patient variation may cause differences in product quality and therapeutic efficacy. Cells respond to their microenvironment, which could alter the composition and concentration of tSVF or the trophic factors secreted by ASCs. For example, during life, (high) glucose blood levels lead to the binding of glucose to ECM in an irreversible and covalent manner, as well as the formation of advanced glycation end products (AGEs). In diabetic patients, due to hyperglycemia, this process is continuous over time, especially when diabetes is not well-regulated and hyperglycemia persists. These AGEs activate bounded cells through the receptor of AGE (RAGE), which has a pro-inflammatory effect which, therefore, also affects the composition of tSVF and the trophic factors of ASCs [36,37]. Smoking and (increases in) physical activity also both negatively influence ASC cell yield and the viability of SVF [38], while debates exist about the influence of factors of age and gender [39,40,41]. Several compounds in cigarette smoke cause increased formation of reactive oxygen species (ROS) in cells. Increased ROS formation causes increased apoptosis, increases inflammation, and disturbs normal physiology. Other practical limitations and legal concerns are the necessity of informed consent of donors to culture their ASCs and the need for clinical-grade culture and lab facilities. Therefore, several methods to obtain SVF out of lipoaspirate have been developed. Either a mechanical or an enzymatic isolation procedure yield tissue SVF (tSVF) or cellular SVF (cSVF), respectively. The mechanical isolation procedure is an easy, non-enzymatic and fast (15 min) intra-operative procedure [42]. In this procedure, ECM and cell–cell, as well as cell–matrix interactions, are preserved. In contrast, during enzymatic isolation procedures, the ECM and all cell–cell/matrix interactions are degraded so as to release to individual cells [42,43]. One of the functions of ECM in tissues including tSVF is to store and release trophic factors upon demand [44]. Thus, the combination of tSVF’s content, including (precursor) ASCs with their secreted trophic factors and ECM, may provide regenerative capacity that exceeds the regenerative potential of cSVF, which lacks the endogenous factors and ECM.

## 3. The Extracellular Matrix as an Instructive Scaffold and Delivery Vehicle for Trophic Factors

Tissue ECM itself is also suitable for therapeutic application because of its physiological function and features. Generally, dry-weight tissue ECM consists of more than 70% collagen. ECMs’ complex structure comprises a mostly interstitial matrix and a small part of the basal membrane consisting of collagen IV, laminin, nidogen, and perlecan. The interstitial matrix is built mostly of constructive force-transducing fibrous proteins such as collagen I (70–85%) and elastin and, in smaller amounts, instructive ECM proteins such as fibronectin and matricellular proteins. Besides proteins, glycosaminoglycans (GAGs) bind and retain water and form a gel in which the protein fibers are embedded. This gel-like matrix is a major shock absorber, cf. cartilage, which is primarily composed of GAGs and proteoglycans (GAGs bound to a protein core) [45,46]. ECM’s characteristics, based on the aforementioned factors, dictate its biomechanical properties such as stiffness, elasticity, and viscoelasticity [43]. These biochemical and mechanical properties together influence cell signal transduction and consequently regulate cell behaviors such as cell growth, proliferation, migration, differentiation, and apoptosis [47,48]. To illustrate, we have shown that ASCs respond to pathological stiffnesses with apoptosis while they thrive in softer matrices [49]. In fact, ASCs secrete MMPs that degrade the matrix in a stiffness-dependent fashion [50]. Thus, ECM fulfills a rather strong instructive function, including direct regulation of relevant physiological processes [51]. In fact, ECM is a dynamic structure that continuously changes, adapts, and (re)acts based on influential factors from surrounding cells, such as the binding of specific surface receptors such as integrin to their recognition sequences in ECM molecules. Not only does ECM act on its surrounding; components contained in ECM e.g., proteoglycans, and GAGs also bind, release and retain trophic factors that are sequestered within ECM, which can be replicated with purified ECM [52]. Over time, or during changes in physiology such as during wound healing, due to diffusion or proteolytic degradation, ECM releases these trophic factors which will activate several pathways in receiving cells [53]. This storage and release function of ECM makes it a potential therapeutic modality. It facilitates the charging of ECM with trophic factors for specific therapeutic goals, such as augmentation of wound healing or remodeling of scar tissue [54,55].

Recent research on ASC-conditioned medium showed that the type of trophic factor has a different release kinetic when bound to an ECM hydrogel [52]. Moreover, release of the trophic factor was dependent on the initial ASC CMe concentration. To illustrate, a more concentrated conditioned medium led to the presence of more inflammation-related cytokines and less pro-regenerative trophic factors. Thus, when choosing a model, the influence of ASC secretome concentration and composition may be major influencing factors.

Eventually, a generic, cell-free, off-the-shelf product would be ideal for wound healing purposes; such a product would have equivalent therapeutic potential as tSVF, ASCs, and ASC secretome (Table 1). Moreover, such a scaffold should meet certain criteria to regenerate dermal tissue. The scaffolds’ structure should be spongy and porous to facilitate metabolite exchange and nutrient supply. Ideally, it should also possess certain mechanical properties, i.e., viscoelasticity and elasticity, similar to dermal tissue. To prevent a host immune response, the scaffold should also lack immunogenicity and should be biodegradable. Ideally, the scaffold should also create a moist environment, since this promotes better wound healing and re-epithelialization. Finally, it also should provide biological cues to influence cell growth and behavior [56,57].

So far, many biomaterials have been proposed to be used for this purpose, such as natural polymers including alginate and chitosan, or synthetic polymers. However, they all have certain adverse properties; alginate and chitosan cannot bind trophic factors and synthetic polymers lack cell adhesion ligands, with both being in contrast to ECM. Therefore, the concept of a decellularized ECM scaffold with incorporated trophic factors has greater potential and superior properties as compared to wound dressings [58]. Decellularization procedures use detergents and salts to isolate ECM from tissue by disrupting cells and washing nuclear content away. Sometimes, mild enzyme digestion is used to make the ECMs’ structure more accessible. Eventually, a DNA- and cellular protein-free ECM powder is yielded, which is converted into a self-gelling hydrogel after mild proteolysis. In decellularized ECM (dECM), the function of the ECM function is preserved, and ECM is still able to bind and store trophic factors and release them sustainedly and gradually over time, which is also the case when charged with the trophic factors of ASCs or ASCs [59]. Besides therapeutic efficacy, another benefit of dECM hydrogels is the off-the-shelf use and storage independence of patients or their (variable) conditions, which contributes to standardizing the product. Other benefits of dECM hydrogels are the feasible standardization of gel manufacturing and less taxing application procedures for patients, independent of the facility. Moreover, the formulation of dECM hydrogels can be adapted to treat several clinical indications, such as patients with heart [60,61], lung [62,63,64,65], or other conditions [66,67]. For some conditions, such as rheumatoid arthritis, charging dECM hydrogels with ASC’s trophic factors would possibly avoid the use of immunosuppressants. For example, in osteomyelitis, antibiotically charged hydrogels would enable local use while systemic side effects are averted.

The feasibility and possible added benefits of trophic factor-loaded dECM hydrogels were recently assessed in rat and mouse models for human dermal damage and summarized in a systematic review [54,55]. It appeared that dECM hydrogels did not inflict an immune response and were considered biocompatible. In some studies, enhanced wound closure and improved angiogenesis were observed after ECM hydrogel treatment, with or without additions of (stem) cells. However, technical and conceptual challenges must be faced in developing a product that also steadily reflects regenerative potency in clinical and histological readouts.

Encapsulating ECM hydrogels with tSVF or whole adipose tissue may also be an interesting approach. The advantages of the former are the immobilization of cells and factors; thus, a longer retention and better therapeutic efficacy can be expected, while for whole adipose tissue, better volume retention is anticipated. The disadvantages are that it concerns a complex product and combined structure which is not off-the-shelf, as well as the fact that the subcutaneously degradability is unknown. Some other examples of the challenges in the manufacturing of ECM hydrogels are the standardization of decellularization methods, the processing and scalability of ECM hydrogel manufacturing, and the preservation of trophic factors [55]. Some efforts have already been made. For example, the decellularization process guidelines already demand that nuclear remnants are removed, the infliction of a host immune response is averted, and the ECM ultrastructure is preserved to maintain its function [68]. Eventually, specific, standardized guidelines need to be developed per clinical indication (Figure 1). Another obstacle is the addition of pre-differentiated induced pluripotent stem cells and progenitor cells such as ASCs, MSCs, or their secretome with hydrogels. Although charging gels with these components has proven to be effective, there exist serious legal obstacles [69]. Some legal issues include obtaining patients’ cells with informed consent for mass production, the serum-free culturing of cells, the enzyme-free production of ECM hydrogels (in decellularization procedures and gel-manufacturing), meeting the requirements of good manufacturing practice (GMP), good laboratory practice (GLP), good clinical practice (GCP), registration as a medical product and the reproducible composition and the quality of the medical product. These challenges warrant extensive collaboration from medical staff, lab technicians, and (inter)national authorities and legislative bodies.

## 4. Conclusions and Future Perspective

This review discussed the therapeutic use of adipose tissue-derived products, tSVF and dECM hydrogels loaded with trophic factors of ASCs, to treat dermal damage. In conclusion, the immune-modulating, pro-angiogenic, and anti-fibrotic properties of tSVF and trophic factors of ASCs may improve scarring based on their characteristics. dECM hydrogels, containing the regenerative potency of tSVF, have proven their feasibility for pre-clinical use. However, technical and conceptual challenges must be faced in developing a product that also has therapeutic efficacy. Future research should focus on product optimization by defining the optimal ECM, tSVF, and ASC trophic factor dosage, standardizing the application and manufacturing of hydrogels, and testing biodegradability or fate in various experimental models. Well-designed clinical trials are warranted to definitively prove the therapeutic regenerative potential of tSVF in scarring and wound healing.

## Figures and Tables

**Figure 1 bioengineering-10-00328-f001:**
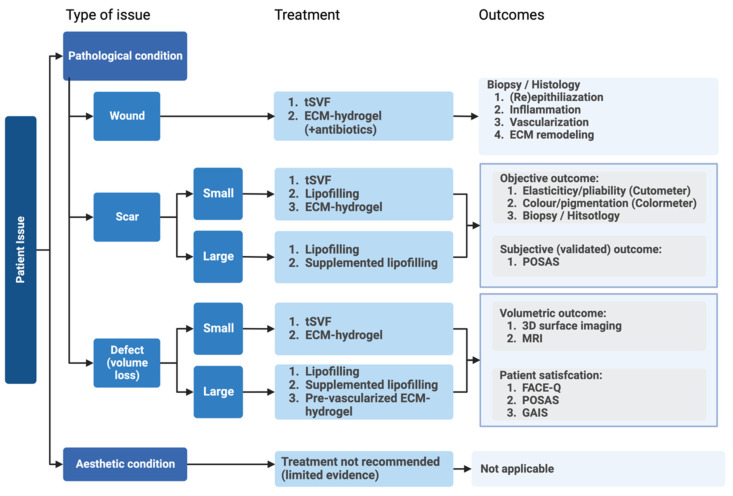
Overview of patient-related dermal issues and the most suitable adipose-tissue-derived treatment options and indicated outcome measures.

**Table 1 bioengineering-10-00328-t001:** Overview of the benefits and disadvantages of adipose tissue-derived treatment options.

Product	Benefits	Disadvantages/Challenges
Lipofilling	Large quantities; easy procedure; limited patient morbidity	Autologous; intra-operative procedure warranted; limited availability slim persons; fat graft retention unclear; influential intra-personal factors; specialized medicine
Supplemented Lipofilling	Large quantities; easy procedure; limited patient morbidity; possibly better fat graft retention	Autologous; intra-operative procedure warranted; limited availability slim persons; fat graft retention unclear; influential intra-personal factors; specialized medicine
Tissue-SVF	Fast, easy, cheap procedure; obtained 1. intra-operatively or 2. from donor tissue; application possibilities for small or medium sizes with high regenerative potential; dose-changeable	Autologous; quantity limited to available adipose tissue; mass cell death after defrosting; specialized medicine
Cellular-SVF	Application possibilities for small or medium sizes with high regenerative potential; dose-changeable	Autologous; expensive, long procedure; chemical use; the need for lab facility, equipment; mass cell death after defrosting; storage; specialized medicine
ASC	Multipotent cell differentiation; high regenerative potential; dose-changeable	Autologous; governmental approval; obtained 1. intra-operatively or 2. from donor tissue or lab culturing; the need for lab facility and equipment; mass cell death after defrosting; storage; specialized medicine
ASC Secretome	Allogenic; dose-changeable; prolonged preservation; various medical indications; no patient morbidity; application possibilities of small, medium, and large sizes	Governmental approval; obtained 1. intra-operatively or 2. from donor tissue or lab culturing; the need for lab facility and equipment; experimental setting
ECM hydrogels	Allogenic; dose-changeable; unlimited preservation; various medical indications; no patient morbidity; application possibilities of small, medium, and large sizes; availability in a large group of patients; specialized medicine and GP	Scalability; governmental approval; indications and therapeutic application not yet determined; experimental setting
ECM hydrogels + ASC secretome	Allogenic; dose-changeable; unlimited preservation; various medical indications; no patient morbidity; application possibilities of small, medium, and large sizes; availability in a large group of patients; specialized medicine and GP	Scalability; governmental approval; Indications and therapeutic application not yet determined; experimental setting

## Data Availability

Not applicable.

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
