# Peer review of "Adipose Tissue-Derived Components: From Cells to Tissue Glue to Treat Dermal Damage"

_bioengineering, 2023, doi:10.3390/bioengineering10030328_

Round 1

Reviewer 1 Report

This review focuses on repair mechanisms in skin disorders and summarizes the many findings that have been known so far. In particular, they devote much of their time to explain the role of the ECM in the subcutaneous tissue, explaining its important and critical functions in the subcutaneous tissue with damage. However, this point of interest is not new and is not much different from what is described in textbooks. Some parts are also mentioned in their previous review: Bioengineering 2018, 5(4), 91; https://doi.org/10.3390/bioengineering5040091. To get the reader's attention, it should be based on a newer perspective.

Table 1 was missing in the peer-review documents. I haven't checked the contents of Table 1, but there are a few references to the secretome in the text. To get a more catchy review, they can focus more on this part and describe its details and roles.

Author Response

Answer:Thank you for your reply and comments on our review. We apologize for the missing figure and table in the submitted manuscript. They have been added in this re-submission. The section ‘Regenerative components of adipose tissue’ mostly concerns ASCs, their secretome and their supposed effects. However, we have added additional information on the ASC secretome at the
end of our manuscript. We hope this answer addresses your remarks.

Reviewer 2 Report

Recommendation: Publish either as is or need minor revisions.

Comments:

This short review is specifically focused on dermal damage treatment by several adipose tissues components including ASCs, SVF and ECM. Although the length is not quite long, authors concisely elaborate on each part and help readers keep up with the technical treads and be inspired in the “future perspective”. While this work has already been organized well and doesn’t have many self-citations, I would only recommend a minor revision on the formatting below.

1. P.1 line 4-13, please be consistent to use “1,2,3… vs. i, ii, iii…” between authors and their affiliations.

2. P.1 line 4, add a comma between “Berend van der Lei, MD, PhD” and “Martin C. Harmsen”.

3. P.1 line 4, please be consistent to use “MD, PhD” vs. “MD and PhD”.

4. P.1 line 9, check the format, is there an extra blank after “&” thus causes “…Pathology &          Medical…”?

5. P.1 line 43, please add the full name of “TGF-β”.

6. P.2 line 64, should be “(v/v)”.

Author Response

Answer:Thank you for your suggestions. We have followed up al remarks and adjusted our text accordingly.

Reviewer 3 Report

This study described the reviews of the regenerative components of adipose tissue. This is an interesting idea and a well-executed review manuscript that has included some meaningful previous studies. I have only one requirement.

Please, add the description and cited some recent studies.

Exp Dermatol. 2022 Dec;31(12):1837-1852./ Stem Cell Rev Rep. 2022 Aug;18(6):1956-1973 / Dermatol Ther. 2020 Nov;33(6):e14277

Author Response

Answer: Thank you for your suggestions for additional publications in the research field. We have added them to our manuscript.

Reviewer 4 Report

This review aims to review the dermal regeneration by adipose tissues components that provide opportunity for building up scaffold and also enhancing wound healing. Lastly, they discuss the disadvantages of applying adipose tissue for therapy. This is a mini review without any highlighted figures reproduced from other findings (I saw the text involved figure 1 but no show in the manuscript). Also, there is no table enlisting the source of adipose tissue, methodology of isolation, therapeutic target/disease model, therapeutic model, outcomes, etc. The authors should also discuss several representative examples of ECM based hydrogels (dynamic/injectable) for the dermal regeneration, especially the adhesive/wearable hydrogel that may encapsulate adipose tissue.

As the authors have mentioned several things about skincare (e.g., scarring), the authors should compare the conventional methods of treating skin wound/scars and application of ASC/hydrogel to improve the healing/treatment. There is a big room for the improvement in this review!

Author Response

Answer: Thank you for your comments on our review.
We apologize for the missing figure and table in the submitted manuscript, it appeared they were uploaded in the wrong section. We hope the inclusion of both the table and figure will clarify and address part of your suggestions for improvement. In Table 1 we focused on the benefits and disadvantages of the therapeutic use and manufacturing of all components we discuss in our review. We did not elaborate extensively on the types of ASC/SVF isolation procedures in this review since we have published a systematic review specifically on this topic a few years ago (we are currently writing an update on the review).1
In Figure 1 we pursue a translational approach and aim to provide clinical perspective; we address which adipose tissue-derived treatment is most suitable per clinical condition. We also propose therapeutic models and outcome measures to encourage clinicians to use valid(ated) outcome measures (as they
are used poorly). Furthermore, the goal of our review was to propose and summarize the therapeutic and regenerative possibilities of adipose tissue-derived components to treat dermal damage, since current treatment options are inadequate and autologous fat grafting has shown benefit in wound
healing, scarring and other dermal conditions. Our goal was not to compare currently used treatments with adipose tissue-derived treatment modalities, although this would also be an excellent idea for a clinical trial or systematic review in the future.
Regarding your comment on the representative examples of dECM hydrogels, we agree this is an interesting and important topic. To investigate the current status of the therapeutic use of acellular matrix to treat dermal damage, we performed a systematic review on this topic. We concluded that only few
animal studies were performed and no human studies. We included the reference (this includes all studies performed on that specific topic) already in our submitted manuscript.2 However, we agree we could have stated this more clearly in our manuscript, we have therefore altered our text to stress this.
Our research group has also performed several animal studies on ECM hydrogels (dermal, cardiac, lung) and we have referred to some of these studies in the manuscript. In one dermal study we used a model where ECM hydrogels were applied under the skin (published), in the other study, hydrogels were
applied on diabetic wounds (under revision).3 We also tested other cardiac models, where hydrogels were injected. We referred to these studies and studies in the field to provide a bigger scope on what has been published recently.
The encapsulation of adipose tissue in hydrogels was also of our very interest. In 2021, we performed a pilot study, where we encapsulated tissue SVF in ECM hydrogels (data not published). The problem we faced, was that the combination of ECM hydrogels and tissue SVF refrained the pre-gels from
gelating, so that the gels did not stay in place. The gels were therefore not suitable for application on wounds, or in great surfaces as it would leak away. Possibly, the encapsulation of adipose tissue or SVF could still be a very good future treatment option if the product is optimized. We expect this would
especially be suitable for for example the restoration of greater amounts of tissue. We agree it would be interesting to also include this information and idea in our review. We thank you for your suggestion and have altered our manuscript.
We hope this answer clarifies and addresses your concerns.

Round 2

Reviewer 1 Report

As this reviewer, I don't think it has improved at all since before.

Author Response

Dear reviewer, 

Thank you for your time to review our manuscript. Your main comment is that our review is not new and is not different as described in textbooks and other publications. A review always contains information that is described by other publications, but our idea is to highlight the importance of extracellular matrix. The critical role of extracellular matrix in regenerative medicine is often underestimated to our opinion. You suggest to focus in this review more on ASC secretome or in other words to write a different review with another perspective. We agree that your idea is of highly interest but was not the primary idea of our study. For that reason, we would like to proceed with the extracellular matrix as main topic for this review. 

Sincerely, 

On behalf of the research group

Dr. J.A. van Dongen 

Reviewer 4 Report

The authors have addressed my concern appropriately.

Author Response

Dear reviewer, 

Thanks for your time and valuable suggestions. 

Sincerely, 

On behalf of our research group, 

Dr. J.A. van Dongen